# Detecting European Aspen (*Populus tremula* L.) in Boreal Forests Using Airborne Hyperspectral and Airborne Laser Scanning Data

**Arto Viinikka** [1,*] , **Pekka Hurskainen** [1,2] , **Sarita Keski-Saari** [3,4], **Sonja Kivinen** [1,3],
**Topi Tanhuanpää** [3,5] , **Janne Mäyrä** [1] , **Laura Poikolainen** [3], **Petteri Vihervaara** [1] and
**Timo Kumpula** [3]

1   Finnish Environment Institute, Latokartanonkaari 11, 00790 Helsinki, Finland;
    pekka.hurskainen@ymparisto.fi (P.H.); Sonja.I.Kivinen@ymparisto.fi (S.K.); janne.mayra@ymparisto.fi (J.M.);
    petteri.vihervaara@ymparisto.fi (P.V.)
2   Earth Change Observation Laboratory, Department of Geosciences and Geography, University of Helsinki,
    P.O. Box 64, FI-00014 Helsinki, Finland
3   Department of Geographical and Historical Studies, University of Eastern Finland, P.O. Box 111,
    FI-80101 Joensuu, Finland; sarita.keski-saari@uef.fi (S.K.-S.); topi.tanhuanpaa@uef.fi (T.T.);
    laura.poikolainen@uef.fi (L.P.); timo.kumpula@uef.fi (T.K.)
4   Department of Environmental and Biological Sciences, University of Eastern Finland, P.O. Box 111,
    FI-80101 Joensuu, Finland
5   Department of Forest Sciences, University of Helsinki, FI-00014 Helsinki, Finland
*   Correspondence: arto.viinikka@ymparisto.fi; Tel.: +358-29-525-1746

**Abstract:** Sustainable forest management increasingly highlights the maintenance of biological diversity and requires up-to-date information on the occurrence and distribution of key ecological features in forest environments. European aspen (*Populus tremula* L.) is one key feature in boreal forests contributing significantly to the biological diversity of boreal forest landscapes. However, due to their sparse and scattered occurrence in northern Europe, the explicit spatial data on aspen remain scarce and incomprehensive, which hampers biodiversity management and conservation efforts. Our objective was to study tree-level discrimination of aspen from other common species in northern boreal forests using airborne high-resolution hyperspectral and airborne laser scanning (ALS) data. The study contained multiple spatial analyses: First, we assessed the role of different spectral wavelengths (455–2500 nm), principal component analysis, and vegetation indices (VI) in tree species classification using two machine learning classifiers—support vector machine (SVM) and random forest (RF). Second, we tested the effect of feature selection for best classification accuracy achievable and third, we identified the most important spectral features to discriminate aspen from the other common tree species. SVM outperformed the RF model, resulting in the highest overall accuracy (OA) of 84% and Kappa value (0.74). The used feature set affected SVM performance little, but for RF, principal component analysis was the best. The most important common VI for deciduous trees contained Conifer Index (CI), Cellulose Absorption Index (CAI), Plant Stress Index 3 (PSI3), and Vogelmann Index 1 (VOG1), whereas Green Ratio (GR), Red Edge Inflection Point (REIP), and Red Well Position (RWP) were specific for aspen. Normalized Difference Red Edge Index (NDRE) and Modified Normalized Difference Index (MND705) were important for coniferous trees. The most important wavelengths for discriminating aspen from other species included reflectance bands of red edge range (724–727 nm) and shortwave infrared (1520–1564 nm and 1684–1706 nm). The highest classification accuracy of 92% (F1-score) for aspen was achieved using the SVM model with mean reflectance values combined with VI, which provides a possibility to produce a spatially explicit map of aspen occurrence that can contribute to biodiversity management and conservation efforts in boreal forests.

**Keywords:** hyperspectral imaging; airborne laser scanning; machine learning; tree species classification; European aspen; boreal forest

## 1. Introduction

Boreal forests are intensively managed in most parts of their range in northern Europe. Forest landscapes are generally composed of pine (*Pinus*) and spruce (*Picea*)-dominated forests with a relatively small proportion of deciduous trees. This is because the forest industry has traditionally favored conifers, and natural disturbances, such as forest fires that support the occurrence of pioneer deciduous species, are nowadays rare [1,2]. Deciduous trees, particularly old trees, have been recognized as an important component of biodiversity in boreal forests [3]. Sustainable forest management increasingly highlights the maintenance of biological diversity [4] and requires up-to-date information on the occurrence and distribution of key ecological features in forest environments.

European aspen (*Populus tremula* L.) (hereafter, 'aspen') is a keystone species in boreal forests that has a sparse and scattered occurrence in northern Europe. Both living and dead aspen trees contribute significantly to the biological diversity of various taxa, such as epiphytic bryophytes and lichens, fungi, invertebrates, and mammals [5,6]. The ecological importance of aspen is illustrated by the fact that numerous aspen-associated species are red-listed [7,8]. Old, large-diameter aspen trees are particularly valuable for species diversity [9,10]. From the perspective of long-term occurrence of aspen-associated biodiversity, the maintenance of a spatial and temporal continuum of aspen trees is a crucial task [5,11,12]. However, the explicit spatial data of aspen occurrence in the boreal landscapes remain scarce and incomprehensive, which hampers biodiversity management and conservation efforts [6].

Remote sensing technology has changed the traditional, relatively expensive and labor-intensive field-based forest inventory to more automated estimations of forest attributes, structure, and tree species [13]. Current operative mapping methods encompassing tree species information in the Nordic countries utilize a whole range of remote sensing data types from aerial RGB and near-infrared images to multispectral satellite data together with LiDAR (Light Detection and Ranging) data [10,14–16]. The favored data source depends on the scale and scope of the final mapping product [17]. The identification of species composition and abundance enables species-level estimates on growing stock and, hence, brings valuable information on timber resources. In addition to this, tree species information acquired by remote sensing is crucial for biodiversity characterization in forest areas [18–21]. Collecting comprehensive reference data, especially for low-number species in the landscape, is challenging. Due to its sparse occurrence, as well as its marginal role in commercial forest management, aspen has often been pooled with other deciduous species in different datasets [22–25]). Thus, there is a demand for aspen discrimination from other species.

The utilization of various hyperspectral instruments has increasingly been studied in tree species detection and classification in different environments (drone: [26–30]; airborne: [31–37]). Hyperspectral remote sensing (also referred to as imaging spectroscopy) provides very high spectral resolution data which consist of hundreds of narrow contiguous spectral bands and have a high signal-to-noise ratio. Different biochemical (e.g., plant pigments and water content) and structural (intercellular spaces and thickness) properties of tree species result in differences in spectral characteristics. Successful discrimination of forest canopy species requires that species have unique and detectable spectral signatures [38]. A variety of different machine learning algorithms have been applied to hyperspectral imagery for tree species classification. In addition to the classification method used, the classification accuracies achieved have a wide variance depending on the spatial and spectral resolution of the data, forest conditions and types, species distribution, and stand structure [36,39–43]. In boreal forest, the use of hyperspectral applications, with the most commonly used machine learning classifiers, support vector machine (SVM) and random forest (RF), has showed high potential for tree species classification [17].

The spectral range of visible and near-infrared (VNIR, 400–1400 nm) and short-wave infrared (SWIR, 1400–2500 nm) can be utilized in tree species classification. Clark et al. [31] examined tree species separability at the crown level using hyperspectral data (400–2500 nm) and found that the most significant wavelengths were spread out over the entire spectral range from the visible to the shortwave infrared. Kalacska et al. [44] used the same data and found that wavelengths with the greatest discriminatory power between trees and lianas were all above 2000 nm. However, increasing the number of features (i.e., bands) in the model increased the importance of near-infrared region bands. Clark et al. [45] showed that vegetation indices, usually ratios of wavelengths representing valuable characteristics of vegetation such as greenness or chlorophyll content, were useful in tree species classification. Hovi et al. [46] conducted a spectral analysis of 25 boreal tree species using a hand spectrometer and found that tree species differed most in the shortwave infrared (1500–2500 nm) and least in the visible (400–700 nm) wavelength range.

In this paper, our overall objective was tree-level discrimination of European aspen (*Populus tremula* L.) from other common tree species (i.e., Scots pine (*Pinus sylvestris* L.), Norway spruce (*Picea abies* (L.) Karst), and birch (*Betula pendula* and *Betula pubescens*)) in the boreal forest landscape, utilizing high-resolution airborne hyperspectral (HS) and airborne laser scanning (ALS) data. Our specific objectives were to:

(1) Compare the performance of different hyperspectral data features in the tree species classification using SVM and RF classifiers, and

(2) Find the most important spectral features to discriminate aspen from the other common tree species.

To answer these questions, we first assessed the performance of different spectral bands (455–2500 nm), principal component analysis (PCA), and vegetation indices (VI) for the best classification accuracy achievable and the robustness of tree species classifications using two machine learning classifiers: SVM and RF. Second, we studied the effect of feature selection for the model performance. Lastly, we investigated the most important spectral features for tree species discrimination. Knowledge of the best-performing classifiers and most important spectral features in aspen discrimination enables us to produce spatially explicit maps of the occurrence and abundance of aspen that can contribute to biodiversity management and conservation efforts in northern boreal forests.

## 2. Materials and Methods

### 2.1. Study Area

The study area (A = 83 km$^2$) is located in the Evo area in the Hämeenlinna municipality in southern Finland (Figure 1). The area includes both managed and protected southern boreal forests [47]. The protected forests include old-growth nature reserves where endangered species are dependent on aspen and coarse woody debris exists. The main tree species are Scots pine (*Pinus sylvestris* L.), Norway spruce (*Picea abies* (L.) Karst), silver birch (*Betula pendula*), and downy birch (*Betula pubescens* L.). European aspen has a relatively sparse and scattered occurrence in the landscape (Figure 2). Agricultural land and built-up areas have a marginal cover in the region. The study area is characterized by a small-scale topographic variation, and most of the region consists of moraine and glaciofluvial deposits. The elevation of the terrain ranges between 125 and 195 m.

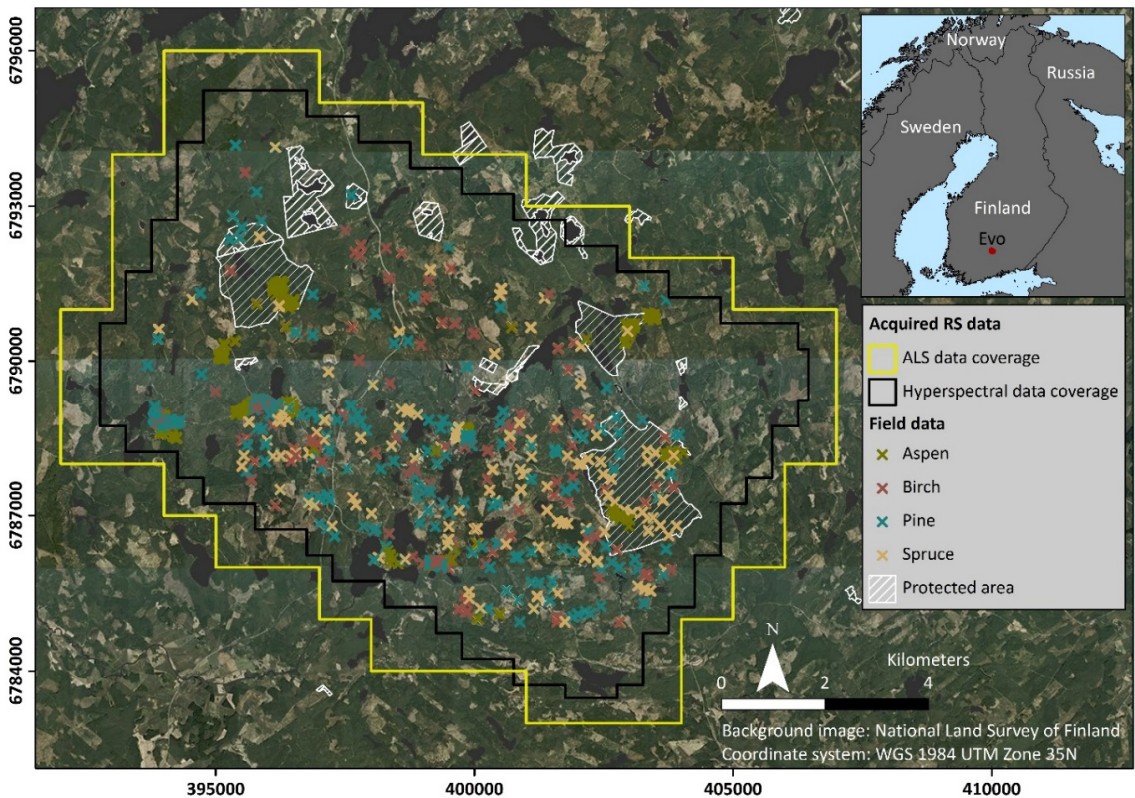

**Figure 1.** Location of study area and acquired remote sensing and field data.

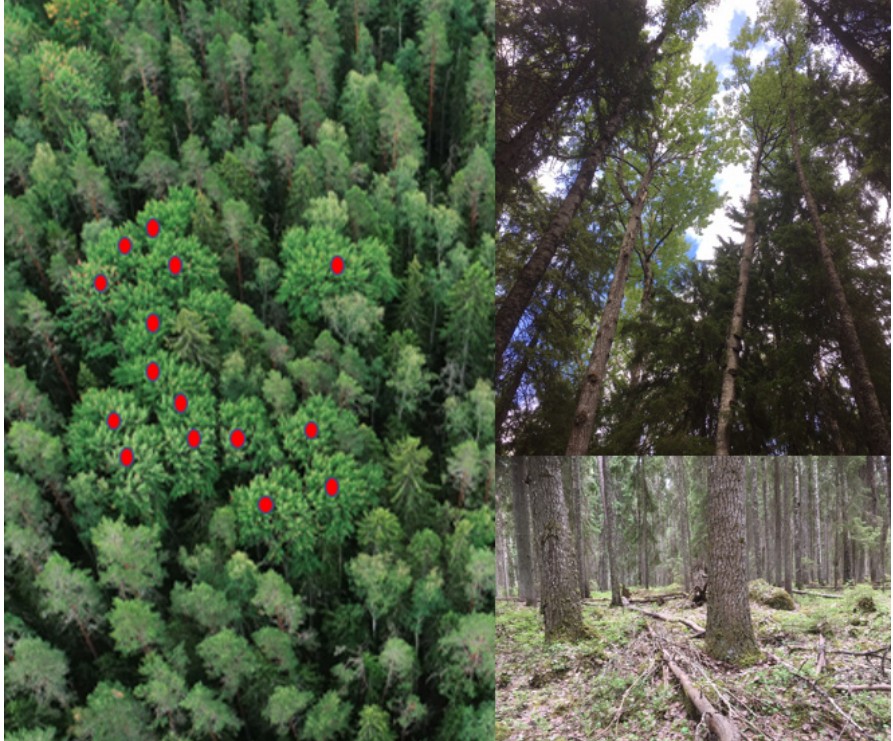

**Figure 2.** On the left, a cluster of aspens (red dots) in old growth forest of Kotinen nature protection area. On the top right, the aspen canopy. On the lower right, the forest ground layer with aspens.

## 2.2. Airborne Hyperspectral and LiDAR Data

The complete analysis workflow is presented in Figure 3. The airborne HS and ALS data were acquired in a flight campaign by TerraTec AS on 16 July 2018 (Table 1).

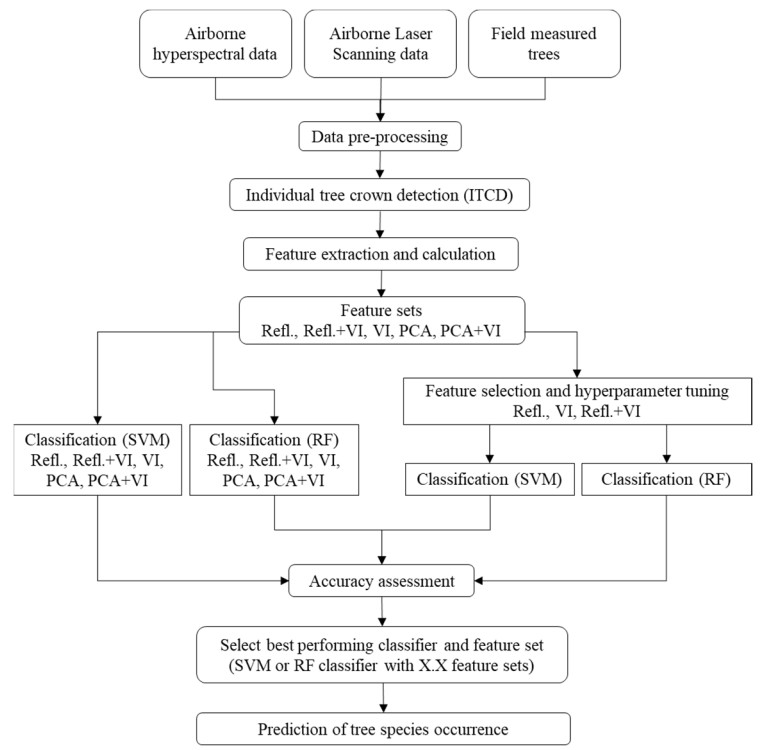

**Figure 3.** A schema of workflow.

**Table 1.** Flight information and sensor parameters.

| Time of Data Capture | 2018.07.16 08:27–11:14 |
|---|---|
| VNIR camera | HySpex 1800–SN00827 |
| VNIR spectral range | 406–995 nm, 186 bands, bandwidth 3.26 nm |
| SWIR camera | HySpex 384me–SN3126 |
| SWIR spectral range | 956–2525 nm, 288 bands, bandwidth 5.45 nm |
| LiDAR scanner | Leica ALS70-HP–SN7204 |
| Pulse density | 10.2 p/m$^2$ |
| Maximum flight altitude | 1500 m above ground level |
| Total imaged area | 82.94 km$^2$ |
| Maximum flight speed | 240.76 km/h |

The preprocessing of the remote sensing data was performed by the data provider. The HS data were georeferenced and orthorectified using *Parge* 3.4 software [48]. Atmospheric correction was done with *ATCOR-4* [49]. The final HS data consisted of georeferenced and orthorectified atmospherically corrected hyperspectral image cubes with a spatial resolution of 0.5 m for the HySpex VNIR-1800 sensor with a spectral range of 405–995 nm, and a spatial resolution of 1.0 m for the HySpex SWIR-384 sensor with a spectral range of 956–2500 nm. SWIR bands above 2453 nm had a poor signal-to-noise level and were masked out from the atmospheric correction and removed from further analyses. The final dataset contained 186 VNIR bands and 274 SWIR bands. Hyperspectral data were divided into 381 square tiles to ease data handling. SWIR data was resampled to 0.5 m resolution for data consistency using the nearest neighbor method.

The ALS point cloud was preprocessed using *TerraPos* software (Terratec, Norway). Automatic methods were used to classify the point cloud into unclassified, ground, and noise classes following the LAS 1.2 standard, and the point cloud subsequently divided into 123 tiles of 1000 m square. The point cloud was further processed into digital elevation models with a spatial resolution of 0.5 m. The digital terrain model (DTM) that represents the bare earth surface was created using only ground points, ignoring vegetation and artificial structures. The digital surface model (DSM) was created from the point cloud using only the single and first of many echoes from the recorded pulses that represent the highest captured surface features such as vegetation or human-made structures like buildings and bridges. The canopy height model (CHM) was derived by subtracting DTM from DSM. CHM represents the actual heights of the vegetation canopy relative to the ground and was later used for individual tree crown detection (ITCD).

### 2.3. Field Data

In order to ensure crown visibility from the hyperspectral data and to facilitate the delineation of old-growth aspens, only the living and standing trees with a diameter at breast height (DBH) of over 15 cm were selected for the analyses. The field data consisted of 6599 individual trees that were compiled from different sources (Table 2). Only the most common species, i.e., Scots pine, Norway spruce, birch, and aspen, were included in the reference data, since the number of trees representing other species was low. During the flight campaign in summer 2018, a hundred circular sample plots with a 9-m radius were measured within the study area. Within each plot, tree species, DBH, tree height, and tree coordinates were recorded for all the trees with DBH >4.5 cm. Furthermore, a total of 300 circular sample plots measured by the Natural Resources Institute Finland in 2018 were available for the study. Plot centers were positioned using the Real Time Kinematic Global Navigation Satellite System (RTK-GNSS). From the center of each plot, the distances and angle for every tree were measured with a high-precision compass and tape measure. Due to the sparse occurrence of aspen in the study area, a total of 599 additional single-tree measurements of aspen were taken using RTK-GNSS. In addition, to standardize the field measurements, complementary measurements for the dominant trees were also acquired.

**Table 2.** Abundance of tree species in field data. Data consisted of trees with diameter at breast height (DBH) of over 15 cm.

| Species Name | Species Count | Species Percentage | Single Tree |
|:---:|:---:|:---:|:---:|
| Scots pine (*Pinus sylvestris* **L.**) | 2570 | 38.9 | 688 |
| Norway spruce (*Picea abies* (**L.**) *Karst*) | 2045 | 31 | 495 |
| Birch (*Betula* **sp.**) * | 1267 | 19.2 | 474 |
| Aspen (*Populus tremula* **L.**) | 717 | 10.9 | 599 |
| **All species** | 6599 | 100 | 2256 |

\* Birch (*Betula* sp.) includes the species downy birch (*Betula pubescens* L.) and silver birch (*Betula pendula*).

### 2.4. Individual Tree Crown Detection

Segmentation was performed using the R-package *itcSegment* [42,50]. Tree tops were located by searching the local maxima within CHM with a moving window method. After the treetops were located, a decision tree method was utilized to aggregate neighboring pixels to the tree crown. To avoid the segmentation of smaller trees (i.e., DBH < 15 cm), a minimum height for the potential local maxima was set to 15 m that corresponds to the 15 cm DBH based on the measured field data. A maximum value of the crown diameter was set to 5 m to capture only the tree top and avoid the spectral signature mixing of the nearby tree crowns.

Field measurements were overlaid with tree segments in order to join the measurements to the corresponding treetop (Figure 4). A total of 3058 (46%) tree measurements that were located inside or less than a 1-m distance from the segments were joined to the segments. Measurements that were

located further away were removed as there was no certainty to which tree these belonged. Some of the larger tree segments contained multiple field measurements within a single polygon. In case the measurements were of the same species, the tree was recorded as that species. Segments containing measurements with multiple tree species were excluded from the analysis.

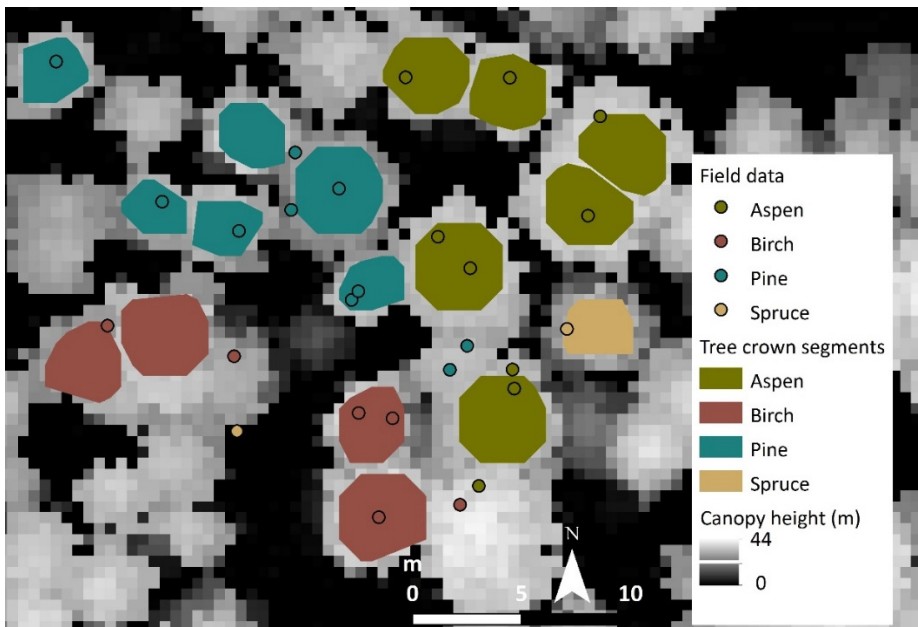

**Figure 4.** An example of the identified tree crown segments according to field measured trees overlaid on top of the canopy height model.

## 2.5. Feature Extraction and Calculation

We compared the performance of mean spectral reflectance, principal component analysis (PCA), and a set of common vegetation indices (VI) for the tree species classification. First, we extracted spectral reflectance of all bands (n = 462) from the pixels intersecting the individual tree segments and calculated the mean spectra inside these segments (Figure 4). Next, we applied PCA to address the problems of high-dimensional data, i.e., redundancy information due to narrow bands located at near-adjacent wavelengths. High dimensionality can decrease the image classification accuracy, which is known as the Hughes phenomenon [51]. PCA was realized using the matrix-based R-package *stats* that reduce data dimensionality while capturing the maximal spectral variation into new uncorrelated components [52,53]. Instead of validating the number of components based on the containing spectral information (e.g., eigenvalue), we tested the performance of all component combinations, starting from the first component until the maximum spectral variation of the original data was included. Using this approach, the components having smaller variation were taken into consideration, which can be important for the better classification result [54].

Lastly, we calculated a set of broadband and narrowband vegetation indices (VI) that have been linked in earlier studies with vegetation structure, biochemistry, and plant physiology in boreal forests (Table S1).

## 2.6. Machine Learning Classification Models

We executed tree species classification both with and without feature selection by applying two machine learning methods from R-package *caret* [55]: the support vector machine (SVM) [56] and the random forest (RF) [57]. They both are non-parametric classification/regression models that are widely used in image classification applying remote sensing data [43,58–60]. We implemented SVM using the linear kernel method that separates different classes by maximizing the minimum margin to the

closest support vector using a linear hyperplane. SVM is a commonly used method due to its ability to generalize well, even with smaller training samples [61]. RF uses a multiple set of decision trees based on a random sample. RF is resilient to overfitting and outliers and can characterize high-dimensional data with many collinear features [57,62,63].

To ensure balanced class proportions for model training and validation, we randomly sampled an equal number of observations (n = 587) for all four species, matching the sample size of the species with the fewest observations (birch, see Table 3). Data partitioning was applied to divide the total of 2348 samples into 70% training and 30% testing sets using proportional stratified random sampling [64]. The first set was used only for model training, while the second set was used as an independent hold-out test set for accuracy assessment, feature importance calculations, and statistical comparisons between different models.

**Table 3.** Field measurements located inside tree segments that was used for the classification.

| Species Name | Tree Count | Training Data (N) | Test Data (N) |
|---|---|---|---|
| Scots pine (*Pinus sylvestris* L.) | 1052 | 406 | 181 |
| Norway spruce (*Picea abies* (L.) *Karst*) | 750 | 406 | 181 |
| Birch (incl. downy birch *Betula pubescens* silver birch *Betula pendula*) | 587 | 406 | 181 |
| Aspen (*Populus tremula* L.) | 611 | 406 | 181 |
| All species | 3025 | 1624 | 724 |

### 2.6.1. Feature Selection

Machine learning algorithms such as SVM and RF are robust to high data dimensionality and many collinear features, but reduction of features is often desirable to avoid model overfitting and to simplify the model [17,62,65]. We tested the effect of feature selection on the classification results by applying recursive feature elimination (RFE), a wrapper feature selection method to simplify both SVM and RF models. The RFE algorithm first fits the model to all predictors and ranks each predictor using its importance to the model. At each iteration, the top-ranked predictors are retained, the least important predictor is eliminated, the model is refit, and the performance is assessed [64]. The models were validated externally with 20-fold cross-validation as an outer resampling method to the RFE algorithm.

### 2.6.2. Hyperparameter Tuning

Instead of accepting default values, the RFE-fitted SVM and RF hyperparameters were tuned in the model training phase to improve classification accuracy with the cost of increased processing time. The hyperparameter tuning was done separately for each model, as the RFE-resulted feature sets used were also different. For RF, two parameters are relevant: the number of trees to be grown (*ntree*) and the number of features used at each node to generate a tree (*mtry*). As suggested by Fox et al. [63], we used a high enough value of *ntree* (3000) to stabilize the results and kept it constant for all models but tuned the *mtry* value separately for each model using an exhaustive grid search with a 10-fold cross-validation repeated 5 times.

In the *caret* implementation for SVM with the linear kernel, only the *cost* (C) parameter, which controls the complexity of the boundary between the support vectors, can be tuned [56,64]. To find the optimal value for C, we used a two-staged grid search approach: first, applying an automatic grid search with 15 values ranging from 0.25 to 4096, and then, a narrower search near to the optimal value found in the first stage.

### 2.6.3. Training, Prediction, and Accuracy Assessment

An optimal number of principal components was selected in three steps: (1) The maximum number of components to be included in the model was selected until the cumulative proportion of spectral variance reached the value 1. This was calculated using the equation:

$$var_c = \frac{sdev^2}{\sum (sdev^2)} \tag{1}$$

where $var_c$ is the cumulative proportion of variance and sdev is the standard deviations of the principal components (i.e., the square roots of the eigenvalues of the covariance matrix). The maximum number of components was 110; (2) the classification performance of principal components was tested by increasing the number of components in the classification until the maximum number of components was reached; (3) components providing the best classification result were selected based on the highest Kappa value and OA.

The performance of the SVM and RF classifiers among the different feature sets were examined using three statistical measures. The macro F1-score considers both precision and recall, equivalent to the user's and producer's accuracy, by calculating their harmonic mean [64]. The values range between 0 and 1, with the best value being 1 and the worst being 0. The F1-score was used to measure the classification accuracy at species-level. Overall accuracy (OA) and Kappa were used to measure the performance of different feature sets. OA refers to the correctly classified trees divided by the total number of sample trees in confusion matrix [66], while the Kappa coefficient compares OA to the classification results that any random classifier would be expected to achieve [67,68]. Thus, Kappa can be used to evaluate and compare the performance of different classifiers. Values range from 0 to 1, and a higher value refers to the better model performance.

We used the McNemar test by applying R-package *stats* to evaluate the statistical significance of the differences between the SVM and RF classification results [53]. The McNemar test statistics were calculated using the total number of correctly and incorrectly classified trees that both classifiers produced [69,70].

### 2.6.4. Model Interpretation

Variable (feature) importance tools are commonly used for enhancing the transparency of "black box" machine learning models, since they try to describe how much a prediction model's accuracy depends on the information of each feature [71]. In the case of SVMs with the linear kernel, it is possible to acquire the coefficients for the hyperplane equations and rank the features by using, for example, the square of the weights as a metric [72]. These coefficients tell which features were weighed the most when constructing the hyperplane. Moreover, because multiclass SVMs are implemented as multiple binary classifiers, either as One vs. One or as One vs. Rest, finding which features matter the most for the full dataset is not a trivial task. For RFs, the most common measurement of feature importance is mean decrease in impurity, which measures how much each feature decreases uncertainty while constructing each decision tree. This method, however, has been shown to be biased towards the features with most variance in the scale or number of features [73,74].

To understand which spectral features are important to discriminate aspen from the other common tree species, we used model class reliance (MCR) proposed by Fisher et al. [71] and defined as "the highest and lowest degree to which any well-performing model within a given class may rely on a variable of interest for prediction accuracy". MCR is based on the random forest permutation feature importance introduced by Breiman [57], but can be applied for any machine learning model and gives a more comprehensive description of importance by considering many prediction models that may fit the data equally well [71].

To calculate MCR, we used the *iml* R-package [75]. We measured the importance of each feature for each model and feature set by calculating the increase in the model's classification error (or loss of

performance) after permuting (shuffling) the feature and repeating the shuffling 20 times to obtain more stable results. A feature can be considered "important" if shuffling its values increases the error, and "unimportant" if shuffling its values leaves the model error unchanged. This is because in the first case, the model relied on the feature for the prediction, and in the latter, because the model ignored it [76]. As the loss function, we used classification error, defined as:

$$ce = \frac{1}{n} \sum_{i=1}^{n} I\left(x_i \neq y_i\right)$$ (2)

where $n$ is the number of observations, $x_i$ is the predicted value for input, $y_i$ is the correct value for input, and $I$ is a function that is 1 when $x_i \neq y_i$ and 0 otherwise.

In this case, MCR is measured as the factor by the model's classification error increases when the feature is shuffled compared to the unpermuted result. Fisher et al. [71] give also practical advice for interpreting the MCR values. For instance, an MCR value of 2 would mean that the model relies heavily on feature X1, in the sense that its loss doubles when X1 is shuffled. An MCR value of 1 signifies no reliance on X1, since the model's loss does not change when X1 is scrambled, while MCR values strictly less than 1 are more difficult to interpret as they rely less on the variable of interest than a random guess.

## 3. Results

### 3.1. Spectral Signatures of the Analyzed Tree Species

Spectral signatures of the extracted tree segments of aspen, birch pine, and spruce are presented in Figure 5. For deciduous species (aspen and birch), the mean spectral reflectance was higher compared to the coniferous species of pine and spruce, especially in the near- and shortwave infrared regions of 750–1400 and 1500–1800 nm. Within the coniferous trees, pine had a higher mean spectral reflectance along these same spectral wavelengths. Relative standard deviation was higher along the whole spectral range for spruce than the other species. For birch, the relative standard deviation was higher in the visible range than for aspen and birch.

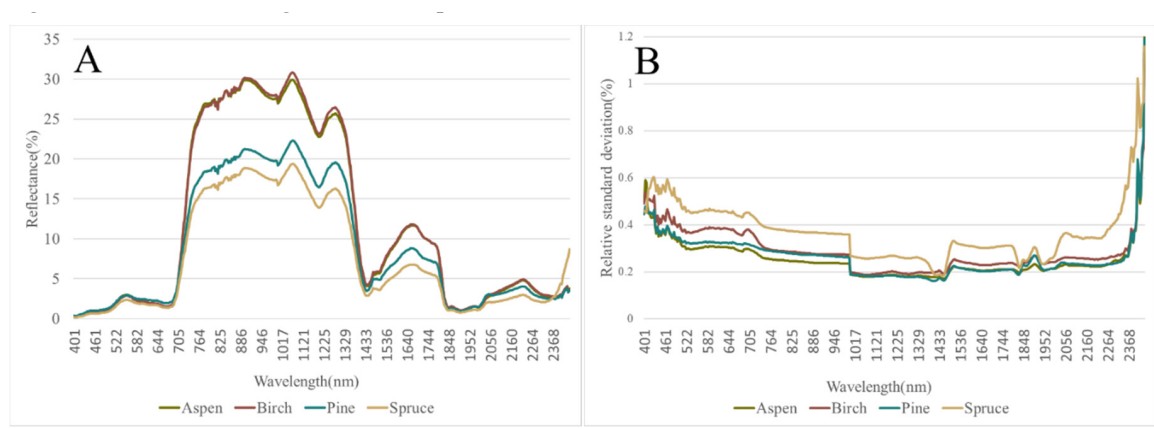

**Figure 5.** Mean spectral reflectance of pixels from the tree segments (**A**). Relative standard deviation of pixels reflectance from the tree segments (**B**).

### 3.2. Impact of the Principal Component Analysis to Classification

The first three principal components contained 91% of the spectral variance, but the classification results remained rather low (Figure 6, Table S2). The highest OA and Kappa were achieved using 38 principal components resulting in values of 0.82 and 0.76, respectively (Table S3). Increasing the number of components in the model above 38 did not improve the model performance.

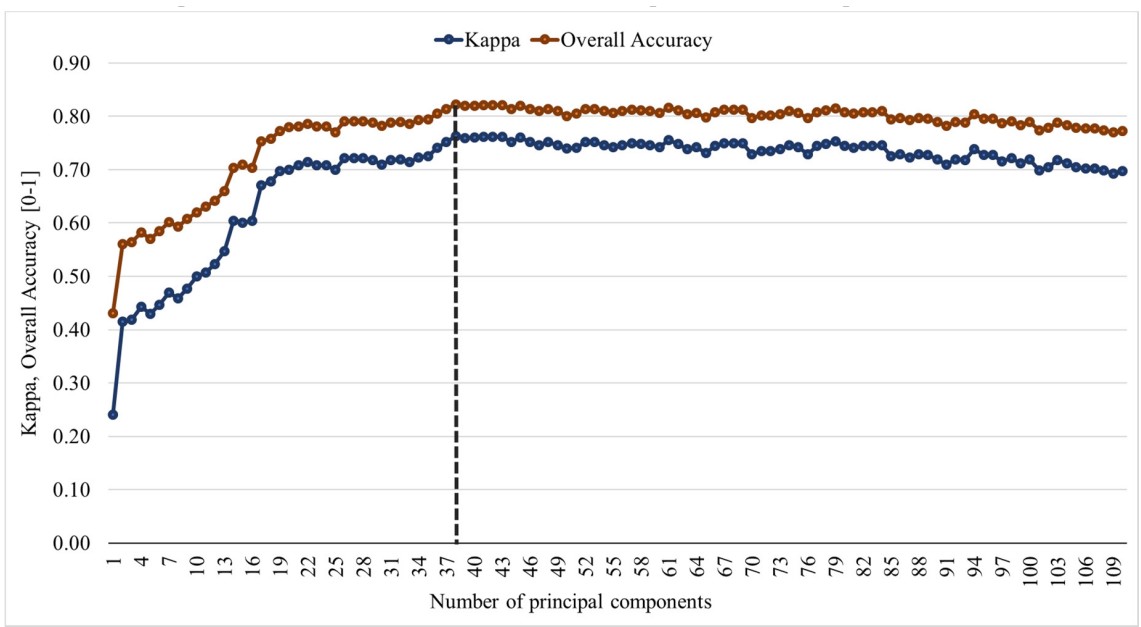

**Figure 6.** Classification results for the different principal component sets using the support vector machine. The black dashed line represents the best classification results according to OA and Kappa with 38 principal components.

### 3.3. Accuracy Assessment and Statistical Comparison of Feature Sets and Models

SVM outperformed RF in tree species classification based on spectral reflectance set with or without VI, according to both the Kappa and OA values (Table 4). Spectral reflectance, VI and their combination did not statistically differ from each other in classification accuracies by the SVM classifier, with an OA of 82–84% and Kappa 0.76–0.78 (Table S4). VI and PCA clearly improved the RF performance, increasing both Kappa (11–16%) and OA (8–11%) compared to the RF results when spectral reflectance was used. The McNemar test did not show a statistical difference between the SVM and RF classifiers when PCA features were used with or without VI (Table S4).

**Table 4.** Classification results for different tree species derived from the confusion matrix without feature selection with different feature sets. RF—random forest; SVM—support vector machine; VI—vegetation indices; PCA—principal component analysis.

| Feature Set | F1-Score | | | | Kappa | Overall Accuracy |
|---|---|---|---|---|---|---|
| | Aspen | Birch | Pine | Spruce | | |
| Reflectance (SVM) | 91% | 82% | 84% | 78% | 0.78 | 84% |
| Reflectance (RF) | 72% | 65% | 74% | 67% | 0.59 | 70% |
| Reflectance + VI (SVM) | 92% | 80% | 82% | 75% | 0.77 | 83% |
| Reflectance + VI (RF) | 82% | 72% | 82% | 76% | 0.71 | 78% |
| VI (SVM) | 89% | 80% | 83% | 77% | 0.76 | 82% |
| VI (RF) | 82% | 73% | 81% | 76% | 0.70 | 78% |
| PCA (SVM) | 91% | 81% | 82% | 74% | 0.76 | 82% |
| PCA (RF) | 88% | 78% | 82% | 75% | 0.75 | 81% |
| PCA + VI (SVM) | 90% | 79% | 83% | 76% | 0.76 | 82% |
| PCA + VI (RF) | 88% | 78% | 81% | 75% | 0.74 | 81% |

Aspen and pine had a higher F1-score than birch or spruce in all feature sets. This indicated a higher mapping accuracy for aspen and pine than for birch or spruce. For aspen, the F1-score varied

from 72% to 92%, and the highest score was achieved with the Reflectance + VI set using SVM. Overall, spruce had the worst classification accuracy.

*3.4. Accuracy Assessment and Statistical Comparison of Feature Sets and Models after Feature Selection*

The accuracy assessment of model performance after feature selection and model tuning is presented in Table 5. Overall, SVM outperformed RF with a statistically significant difference, regardless of which feature set was used (Table S4). With RF, the highest OA and Kappa metrics (77% and 0.69) were achieved with feature sets of either VI or both reflectance and VI, while all three SVM models gave almost identical results (OA 83–84% and Kappa 0.77–0.78). Among the three SVM models, no significant difference was found, whereas the RF reflectance feature set had a statistically significant difference in comparison to the other two RF models (Table S4).

The same general pattern was observed when comparing the F1-scores of individual tree species: SVM produced very similar results regardless of the fitted feature set, with a maximum difference of 1 percentage point between the feature sets. For the RF models, fitted feature sets made a big difference: the reflectance feature set produced the least accurate classification, which was significantly improved when VI alone or both reflectance and VI were used.

Aspen was classified with the highest accuracy (F1-score) of 91% with the SVM model fitted with both VI and reflectance. The same feature set (although with a considerably smaller number of features) worked best when using RF to classify aspen, with an F1-score of 82%.

**Table 5.** Classification results for models after feature selection was applied. K—Kappa; OA—overall accuracy; RF—random forest; SVM—support vector machine; VI—vegetation indices; PCA—principal component analysis.

| Feature Set | Features | F1-Score | | | | Kappa | Overall Accuracy |
|---|---|---|---|---|---|---|---|
| | | Aspen | Birch | Pine | Spruce | | |
| Reflectance (SVM) | 370 | 90% | 80% | 87% | 75% | 0.77 | 83% |
| Reflectance (RF) | 144 | 77% | 69% | 76% | 62% | 0.61 | 71% |
| VI (SVM) | 44 | 90% | 80% | 86% | 76% | 0.77 | 83% |
| VI (RF) | 43 | 81% | 74% | 82% | 70% | 0.69 | 77% |
| Reflectance + VI (SVM) | 290 | 91% | 81% | 86% | 75% | 0.78 | 84% |
| Reflectance + VI (RF) | 37 | 82% | 75% | 81% | 70% | 0.69 | 77% |

The confusion matrix of the highest OA, Kappa, and aspen F1-score for both the SVM and RF models, fitted with the combination of VI and reflectance, is presented in Figure 7. In both models, aspen is mostly mixed with birch, whereas pine and birch are mostly mixed with spruce.

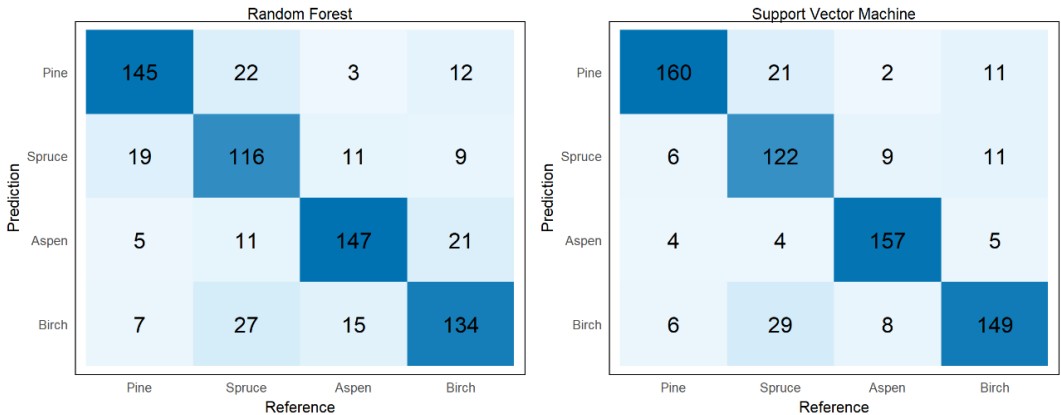

**Figure 7.** Confusion matrix for the best SVM and RF models fitted with reflectance values and VI after feature selection.

### 3.4.1. Impact of the Feature Selection and Model Tuning to Classification

The results of the feature selection and its impact on cross-validated classification accuracy, as well as the optimal number of features leading to the best accuracy, are presented in Figure 8 and Table 5. The feature selection algorithm was successful in considerably simplifying all models except the ones trained with only vegetation indices, where two features and one feature were removed for the RF and SVM models, respectively.

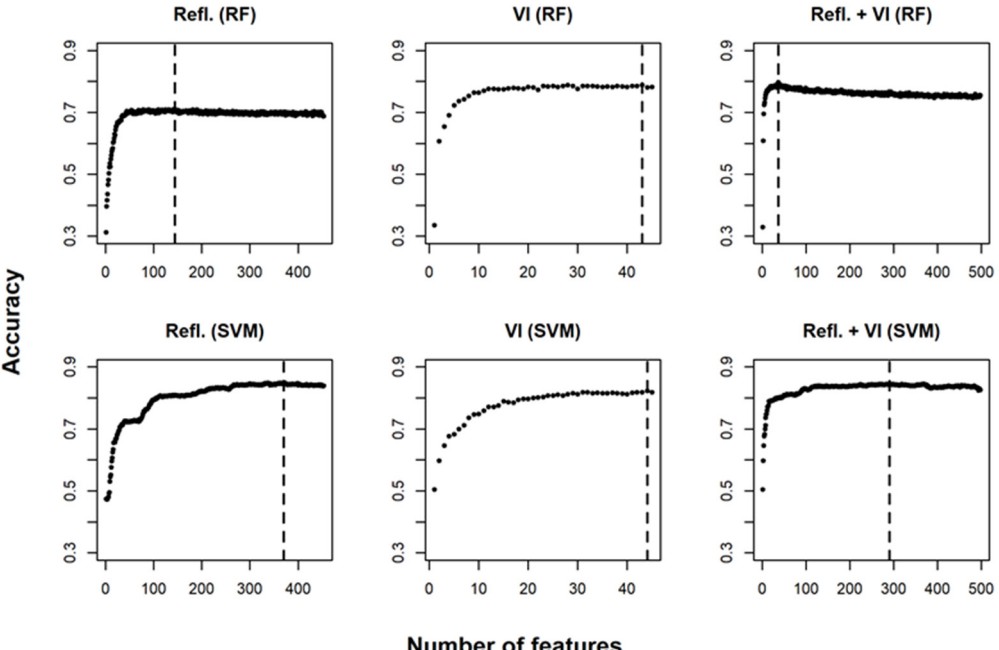

**Figure 8.** RF and SVM classification accuracy (cross-validation) versus number of features (feature subset size) selected at each step of the RFE. The dashed vertical line denotes the numerically best subset size, leading to best accuracy.

The largest number of features for both RF (n = 144) and SVM (n = 370) were kept for models trained with VNIR and SWIR mean reflectance, and the feature count was reduced by 68% and 18%, respectively. Models trained with reflectance and VI were the most parsimonious after feature selection: 37 features for RF and 290 for SVM were kept, reducing the number of features by 93% and 42%, respectively.

After an exhaustive grid search, the optimized mtry values for the three RF models were 51, 15, and 9. For SVM models, the optimized hyperparameter values for C were 0.6, 0.55, and 2.

### 3.4.2. Feature Importance

The permutation feature importance (model class reliance) plots for SVM models (Figure 9) and RF (Figure 10) revealed several interesting insights for the interpretation of the classification decisions. The F1-scores for the tree species of the SVM models were always markedly higher than those of the RF models (Table 5). Thus, it is possible to identify important features (classification error of greater than 2) from all SVM feature sets for aspen, birch, and pine, but not for spruce (Figure 9). In contrast, for the RF models, all features had a classification error of less than 1.4, and the differences between the errors were much smaller, which makes model interpretation much more difficult than for SVM (Figure 10).

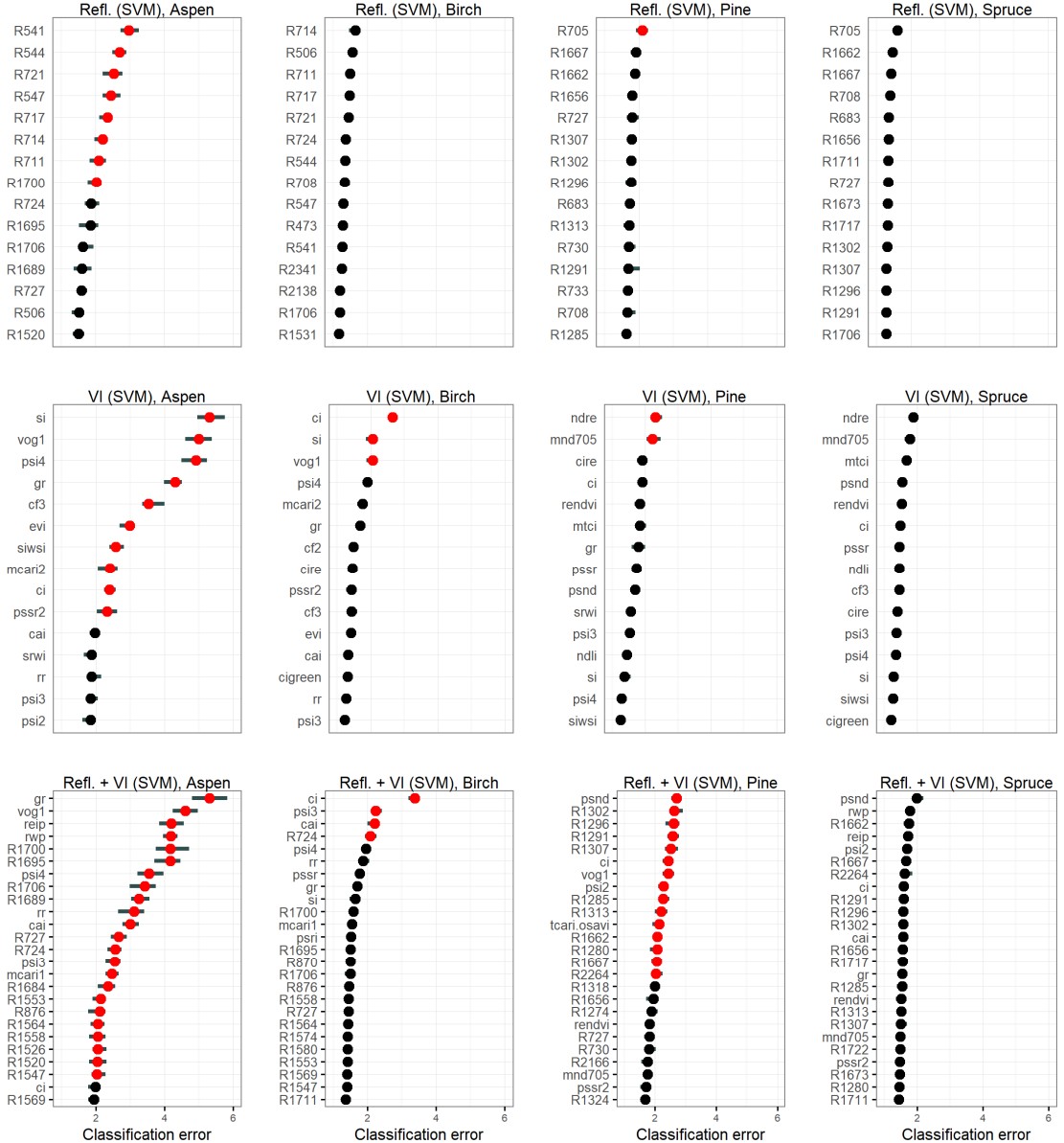

**Figure 9.** Permutation feature importance for classification of the four tree species applying the Support Vector Machine (SVM) model. The models were fitted with reflectance values (top row, 15 most important features), vegetation indices (VI) (middle row, 15 most important features), and combined reflectance values and VI (bottom row, 25 most important features). The importance is measured as the factor by which the model's classification error increases compared to the unpermuted classifications when the feature is permuted. Each permutation was repeated 20 times. Here, the horizontal line denotes the 5% and 95% quantiles of importance values, the points denote median importance, and red color indicates that median classification error is above 2, suggesting heavy model reliance on the feature.

We found important common features for the discrimination of deciduous trees (aspen, birch) from coniferous trees (pine, spruce) with the SVM model. The Cellulose Absorption Index (CAI), Plant Stress Index 3 (PSI3) and red edge (724 nm) were found to be important for deciduous trees and the Pigment Sensitive Normalized Difference (PSND) was an important feature for the coniferous trees when the model was fitted with the combined reflectance values and VI (Figure 8, bottom row). Interestingly, the SVM model fitted only with VI, the most important features to discriminate between deciduous and coniferous trees differed comparing to the combined reflectance and VI (Figure 8, middle row). The Spruce Index (SI), Vogelmann Index 1 (VOG1), and Conifer Index (CI) were

important for both aspen and birch in the VI model. The classification of coniferous trees benefitted from the Normalized Difference Red Edge Index (NDRE) and Modified Normalized Difference Index (MND705), although with low classification error for spruce. In addition, for the discrimination of pine, reflectance wavelength ranges in shortwave infrared (1280–1313, 1662–1667, and 2264 nm), and five other vegetation indices (PSND, CI, VOG1, Plant Stress Index 2 (PSI2), and Transformed Chlorophyll Ratio Index/Optimization of Soil-adjusted Vegetation Index (TCARI/OSAVI)) were important (Figure 9, bottom row).

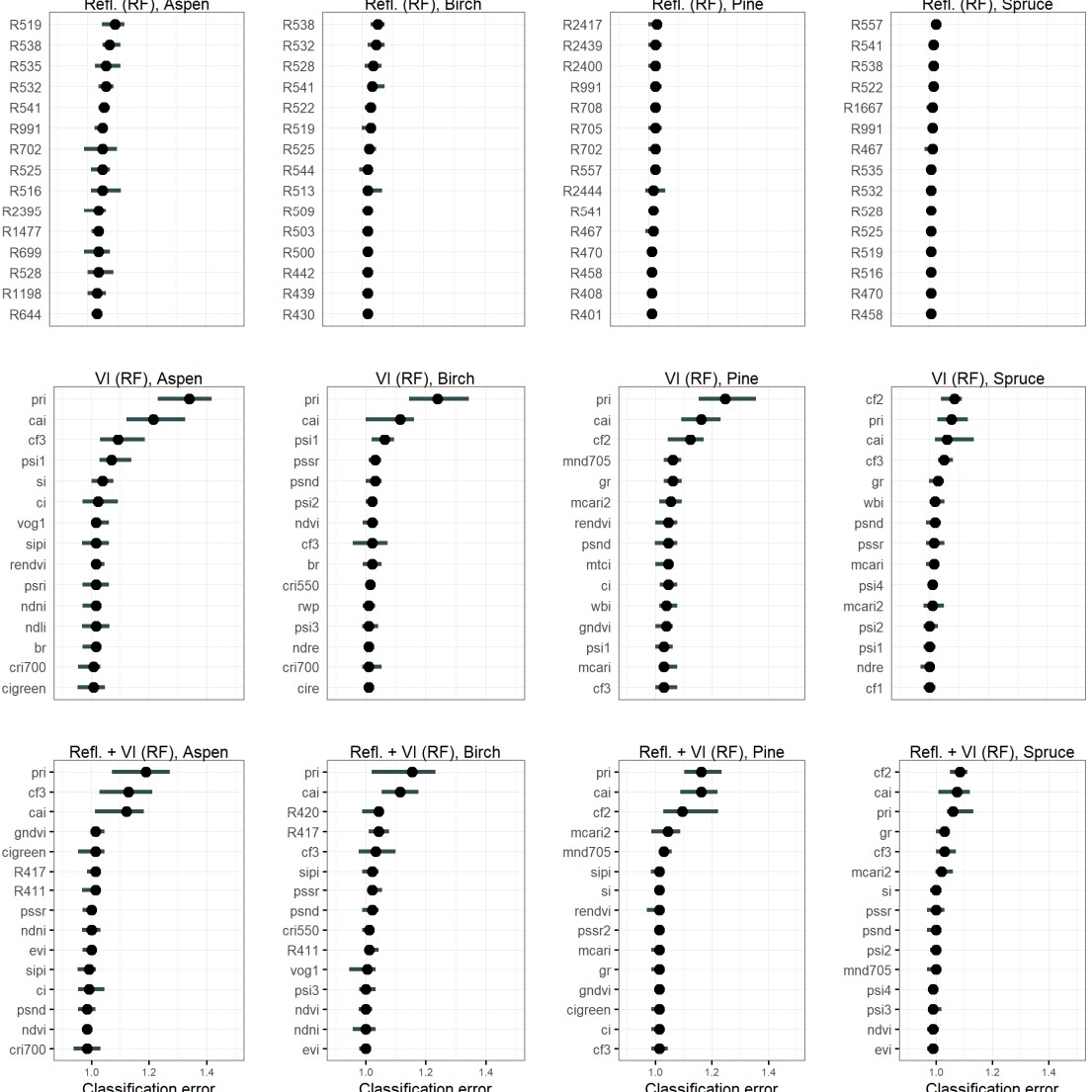

**Figure 10.** Permutation feature importance of the fifteen most important features for classification of the four tree species applying the Random Forest (RF) model fitted with reflectance values (top row), vegetation indices (VI) (middle row), and combined reflectance values and VI (bottom row). The importance is measured as the factor by which the model's classification error increases compared to the unpermuted classifications when the feature is permuted. Each permutation was repeated 20 times. Here, the horizontal line denotes the 5% and 95% quantiles of importance values, the points denote median importance, and red color indicates that median classification error is above 2, suggesting heavy model reliance on the feature.

Even though the classification error remained rather low in all feature sets in RF models, four VI stood out slightly more important than the others between all the four tree species (Figure 10). CAI and

the Photochemical Reflectance Index (PRI) were found in top 3 indices for all tree species in RF. Similarly, Chlorophyll Fluorescence $_{R761/R757}$ (CF3) showed to be somewhat important for deciduous trees, and Chlorophyll Fluorescence $_{R690/R630}$ (CF2) for coniferous trees.

In the SVM models, the higher the F1-score of a tree species was, the larger the number of features were assessed to be particularly important. Since aspen had the highest F1-score in all models, particularly important features were many, and they were easier to identify. In contrast, spruce was the most difficult species to classify, which reflects in the lowest F1-score, and consequently, not a single feature stood out as being particularly more important than the others.

The best performing SVM model fitted with the combined reflectance values and VI relied heavily on 23 important features for the discrimination of aspen from the other three species (Figure 9). Out of these, nine were VI, such as Green Ratio (GR), VOG1, Red Edge Inflation Point (REIP), and Red Well Position (RWP), all having a classification error of greater than 4, while for the other five indices (Plant Stress Index 4 (PSI4), Red Ratio (RR), CAI, PSI3, and Modified Chlorophyll Absorption in Reflectance Index 1 (MCARI1)), the error was between 2 and 4. Fourteen out of the 23 important features were reflectance bands from shortwave infrared (1520–1564 and 1684–1706 nm), red edge range (724–727 nm), and near-infrared (876 nm) wavelength ranges. When considering also the wavelengths incorporated in the algebraic equations of the VI (Table S1), important features for aspen discrimination were found in different parts of the full electromagnetic spectrum, excluding the blue region.

The variance of vegetation indices based on the spectral properties of the field measured trees are presented in Figure 11. Most of the indices showed a clear difference in the spectral signature between deciduous trees (aspen, birch) and coniferous trees (pine, spruce). Within the functional classes, the spectral difference is much less obvious.

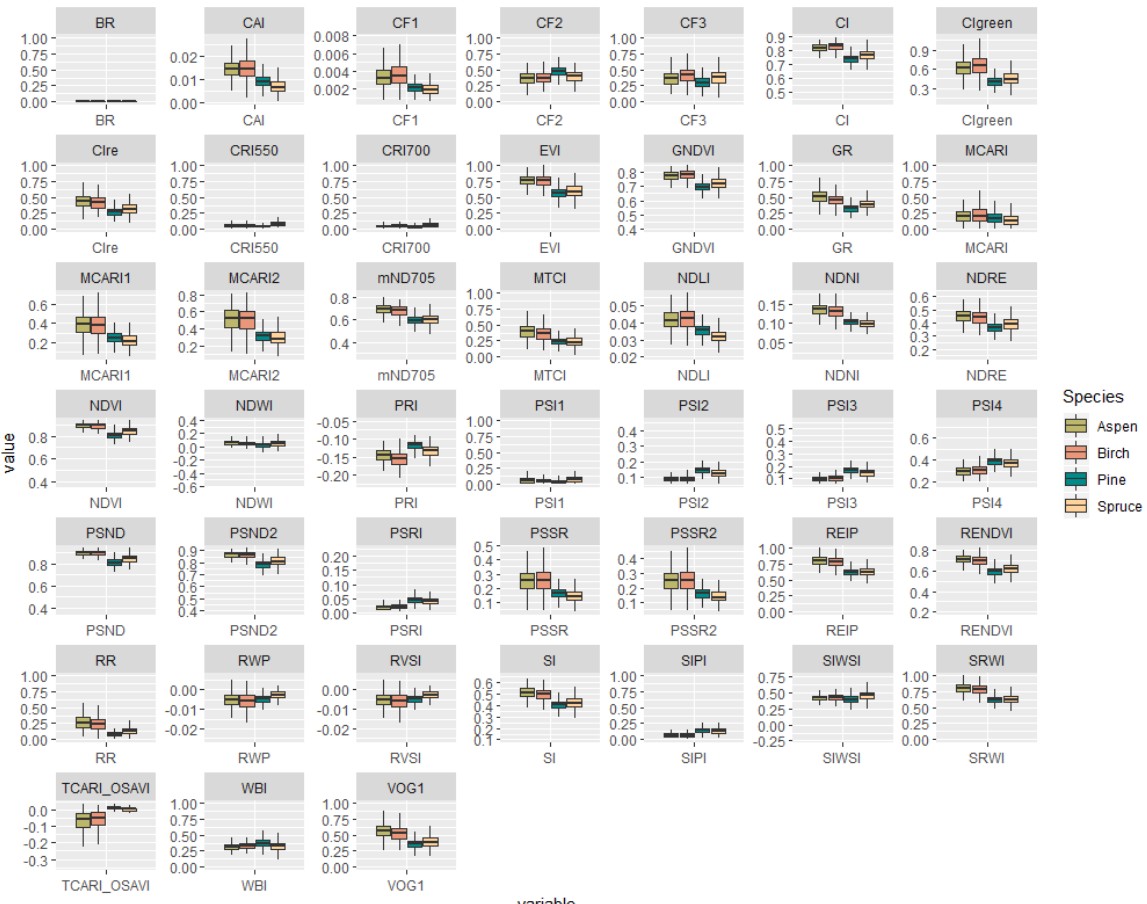

**Figure 11.** Variation of vegetation indices in different tree species.

## 4. Discussion

### 4.1. Impact of Classifiers, Features, Feature Selection, and Segmentation

The overall tree classification accuracy of aspen, birch, pine, and spruce in this study was relatively good, with Kappa varying from 0.59 to 0.78 and OA from 70% to 84%. At the species level, both SVM and RF were able to classify aspen and pine with a higher accuracy (i.e., higher F1-score) compared to birch and spruce, regardless of the feature set used. The comparison of classification accuracies among studies is challenging due to different tree species, forest structure, the spatial and spectral scale of the hyperspectral data, analysis methods as well as the number of species to detect, as noted also by Waser et al. [77]. Additionally, the abundance and spatial distribution of the species affects the species classification, resulting in differences in the classification accuracy among vegetation types [78]. However, whereas spectroscopic measurements of dry leaves at the leaf-scale can lead to very high discrimination accuracies, with a Kappa of 0.96 for 19 plant species representing different taxonomic families and vegetation types [79], our results are comparable to other canopy-level studies, where variation in spectral imagery is higher due to the changing conditions of sunlight illuminance and shadow effects. Dalponte et al. [33] used hyperspectral data together with airborne laser scanning data to discriminate pine and spruce from the deciduous trees in boreal forest using support vector machine classifier and resulting in an overall accuracy of 88% and Kappa of 0.79. However, the deciduous species were always poorly classified (producer's accuracy less than 50%), which was explained by the fact that the automated delineations detected fewer broadleaved trees as pine was the dominant species in the area. This is a common problem in remote sensing that can cause unbalanced training and testing sets among the species that can lead to data-driven errors in classification accuracy [43]. In our study, we used equal sample size to exclude the potential data-driven errors as we wanted to focus only on classification differences originated from the species' spectral properties. In a more diverse species discrimination study, Tuominen et al. [28] reported similar accuracies (overall accuracy of 82%) for the discrimination of 26 tree species in Finland but only when using the combination of mean reflectance values ranging from 400 to 1600 nm and 3D point cloud features. Without the point cloud features, the accuracy was 11% lower comparing to our results. Even though this can partly be explained by the lower number of tree species in our study, the inclusion of the shortwave infrared range may have contributed to the classification accuracy, as can be seen from the number of important wavelengths and VI in the SWIR range. Similar findings, regarding the importance of these particular regions were reported in the review by Fassnacht et al. [17].

Earlier studies have shown that both SVM and RF classifiers perform equally well in tree species classification [80,81]. In our results, SVM models outperformed RF in all feature sets, having 4–14% higher OA and 0.06–0.09 higher Kappa, except when PCA features were used. SVM performed equally well despite the used feature set, while PCA clearly enhanced the RF ability to discriminate between all four tree species. Although most of the spectral variation was concentrated to the first three principal components (91%), the best OA and Kappa values required 38 components when classification was performed using PCA features only. This emphasizes the importance of components containing a very small spectral variation (38 principal components had a spectral variance of 0.0000894%) in tree species classification with narrow band hyperspectral data and supports the suggestion made in Björklund [82] to avoid simple rules of thumb such as eigenvalues larger than 1.0 and loadings larger than 0.5 when selecting the optimal number of principal components for the model.

Feature selection did not have a significant impact on OA and Kappa values, but it was possible to achieve the same level of accuracy with a considerably smaller number of features comparing to classification results with full feature sets. In the best-performing SVM model, the number of features decreased by 42%, and likewise, by 93% in the best-performing RF model. This is beneficial in two ways: first, the interpretation of the important features driving the discrimination of tree species becomes easier as the most irrelevant features have been removed, and secondly, a more parsimonious

model is more likely to find a better balance between fitting the training data well and avoiding overfitting, allowing a more general application of the model [17].

The reflectance in the visible range is affected the most by pigment content, with the green hump between the blue and red regions of high chlorophyll absorbance [83]. The area and height of the green hump is inversely related to the pigment contents of chlorophyll, carotenoids, and anthocyanins. Particularly, leaf chlorophyll content affects the positioning of the red edge, with a shift to higher wavelengths due to a higher chlorophyll content, as shown by both experimental and simulated data [84–86]. The chlorophyll content of deciduous tree leaves is generally higher than that of coniferous needles [87]. Thus, in our study, the location of the red edge inflection point (REIP) was in higher wavelengths in deciduous trees comparing to coniferous trees (Figure 11). In the NIR and SWIR range, deciduous and coniferous species are known to have a level difference in reflectance, with deciduous trees having a constantly higher reflectance [83,88], which was also the case in our study. On the leaf level, reflectance in the NIR range is affected predominantly by the leaf structure, such as the ratio of mesophyll cell surface area to intercellular air spaces [83]. However, on the canopy level, the lower level of reflectance of conifers in the NIR range may be affected by forest canopy structure, such as self-shading, visibility of the trunk, clumping of the needles as well as the scattering and recollision properties of the canopy [83].

The high classification accuracy for the individual tree species (i.e., F1-score) coincided with the number of important features in the model. For example, aspen and pine that were most accurately classified with the SVM model benefitted from the important features more comparing to birch and spruce. We identified seven vegetation indices that were important for the discrimination of deciduous and coniferous trees in the SVM model, either in the vegetation index set or combined reflectance and vegetation index feature set. The Cellulose Absorption Index (CAI), Plant Stress Index 3 (PSI3), Spruce Index (SI), Vogelmann Index 1 (VOG1), and Conifer Index (CI) were the important ones for deciduous trees, whereas the Normalized Difference Red Edge Index (NDRE) and Modified Normalized Difference Index (MND705) are important for coniferous trees. The indices are commonly based on specific absorption features that lead to lower reflectance values on the specific wavelengths. For example, CAI is based on an absorption feature common to cellulose and lignin on wavelengths 2019 and 2206 nm, and it was originally used to detect residual plant material on an agricultural field [89]. In our study, the difference in CAI between deciduous and coniferous trees was probably due to the higher lignin content of the latter. SI and CI were proposed to have higher values for spruce and coniferous trees, respectively, using at-sensor radiance data of birch, pine, and spruce [90]. However, we failed to reestablish the result using atmospherically corrected hyperspectral data. Instead, deciduous trees had higher values for SI and CI in our data. VOG1, PSI3, and the indices associated with the conifers, NDRE and MND705, were calculated based on the reflectance near the red edge inflection point. Similarly, the spectral reflectance at 724 nm was found to be important in discriminating deciduous and coniferous trees. The red edge range, and indices based on it, such as PSI3, are also known to respond to several plant stress factors [91]. MND705 is especially good in predicting chlorophyll content in coniferous needles, since it accounts for high specular reflectance on the needle surface [92]. Thus, the indices used in chlorophyll content estimation had high importance in discrimination between deciduous and coniferous trees in this study.

CAI was one of the most important VI not only in SVM but also in the RF model. The other ones for RF, Photochemical Reflectance Index (PRI), Chlorophyll Fluorescence R761/R757 (CF3), and Chlorophyll Fluorescence R690/R630 (CF2), estimate photosynthesis efficiency (Sims and Gamon [92]) and chlorophyll fluorescence (Zarco-Tejada [93]), respectively.

The positional accuracy was low for some trees that were measured using circular sample plots. Removing all measurements that were located over 1-m distance from the segments helped, but still, some of the field measurements might have been matched with the wrong tree crown. In our approach, we performed individual tree detection and tree species classification separately based on different data. There may have been some cases where LiDAR and hyperspectral data were not perfectly aligned

that may have resulted in segments not containing the correct tree pixels of the hyperspectral data, leading to potential classification errors. Because of this possible misalignment, it is reasonable for future studies to consider methods that can perform detection and classification from a single image in one single step, outputting an image with each pixel labeled (semantic segmentation). Modern deep learning methods, such as U-Net [94], have shown good results in various semantic segmentation tasks, ranging from biomedical to geospatial domains. However, these methods require segmented masks as their training data and according to our knowledge, this has not yet been applied to hyperspectral imagery. Nevertheless, utilizing deep learning methods either for combined detection and classification or just plain classification is an interesting research direction in the future.

### 4.2. Impact of Spectral Features for Aspen Discrimination

Of all the species used in our study, aspen resulted in the highest accuracy (F1-score 92%) when the SVM model was used with the combination of reflectance values and VI. This is noteworthy, since rare and sparse species are commonly misclassified [78]. With feature selection, it was possible to decrease the number of used features by 42% without significantly decreasing the high level of accuracy (F1-score 91%, Table 5). In previous studies, the accuracy for European aspen detection with various remote sensing methods has been rather low comparing to other dominant species. In the latest aspen review, Kivinen et al. [6] reported a user's accuracy ranging from 56% to 86% and producer's accuracy from 24% to 71% for aspen. In this sense, our results for aspen discrimination were relatively good.

The higher accuracy for aspen comparing to other species in this study was probably due to the higher number of important features specific to aspen. Of the vegetation indices, Green Ratio (GR), Vogelmann Index 1 (VOG1), and Plant Stress Index 4 (PSI4) were important for both feature sets: VI or the combination of VI and reflectance (Figure 9). One reason for that might be the higher mean values of these particular indices for aspen than the other species (Figure 11). While VOG1 and PSI4 were important for the deciduous trees, the Green Ratio (GR) showed to be more important for aspen than birch. GR was originally developed by Waser et al. [77] to monitor ash mortality due to fungal pathogens, but they noted that it was useful also in species classification, which was the case also in our study. In addition, the red edge inflection point (REIP) and Red Well Position (RWP) showed to be important for aspen discrimination. REIP and RWP are related to the shape differences in the spectrum in positioning of the red edge and the minimum reflectance of the red range, respectively [95].

The most important wavelengths to discriminate aspen were the reflectance bands of the red edge range (724–727 nm), near-infrared (876 nm), and shortwave infrared (1520–1564 and 1684–1706 nm) (Figure 9). The wavelengths 1500 and 1520 have been shown to correlate with cellulose content and leaf thickness [79,96]. Lignin and cellulose content are generally higher in coniferous needles than deciduous leaves [97]. A higher content in needles leads to higher absorbance and therefore, a lower reflectance on those specific wavelengths. The wavelengths 1650–1700 nm have been shown to be related to phenolic compounds in purified standard compounds (condensed tannins, salicortin, and tremulacin) as well as in spectroscopic measurements of trembling aspen (*P. tremuloides*) and paper birch (*B. papyrifera*) leaves [98]. Similarly, near-infrared reflectance spectroscopy of trembling aspen and paper birch leaves showed features at 1685–1705 nm associated with phenolic compounds [99]). A narrow reflectance feature at 1660 nm was found to be specific to most phenolic compounds, whereas lignin had a broad feature at 1670 nm [100]. Therefore, the spectral differences between aspen and the other species in this study on those wavelengths imply that phenolic compounds in aspen leaves contributed to the canopy reflectance of aspen.

Aspen was mostly mixed with birch and spruce in the species classification (Figure 7). For spruce, the one explanation is the higher relative standard deviation of the reflectance along the whole spectral range causing it to get mixed with other species (Figure 5b). In general, spruce stands are found on richer soil compared to pine and therefore, can be surrounded by deciduous trees such as aspen and especially birch as admixture [101]. Thus, the sensor that recorded the spectral reflectance from spruce might have originated partly from aspen and birch, especially in the boundary surface between

the neighboring tree segments. Misclassification of aspen as birch or other deciduous tree species due to the similarity of the tree structure and high chlorophyll content is acknowledged in previous studies [65,80]. This study strengthened that observation by highlighting the PSI3, SI, VOG1, and CI mutually important VI for both aspen and birch. In addition, birch did not have any features that would not have been important also in aspen classification that might have influenced misclassification between aspen and birch (Figure 9).

### 4.3. Considerations for Cost-Effectiveness and Upscaling the Results

Hyperspectral data together with ALS provided a possibility to produce a spatially explicit map of aspen occurrence that can help to target local-scale biodiversity management and for planning the conservation efforts in forest areas (Figure 12). The main limitations of applying the proposed approach to larger scales more relevant for forest biodiversity and conservation management are the limited spatial coverage and high data acquisition and processing costs associated with airborne hyperspectral and LiDAR data collection. However, from a management perspective, the accurate delineation and classification of every single tree in a forest is not as important as delineation and classification of homogeneous tree stands, or area-based estimation of species occurrences which does not require such high-resolution data. In this study, we have identified the most important spectral features for aspen and other dominant species in Finnish boreal forests. This valuable information can be used in future tasks utilizing data collected by satellite sensors, contributing to conservation prioritization as a part of the national-scale conservation planning task [102].

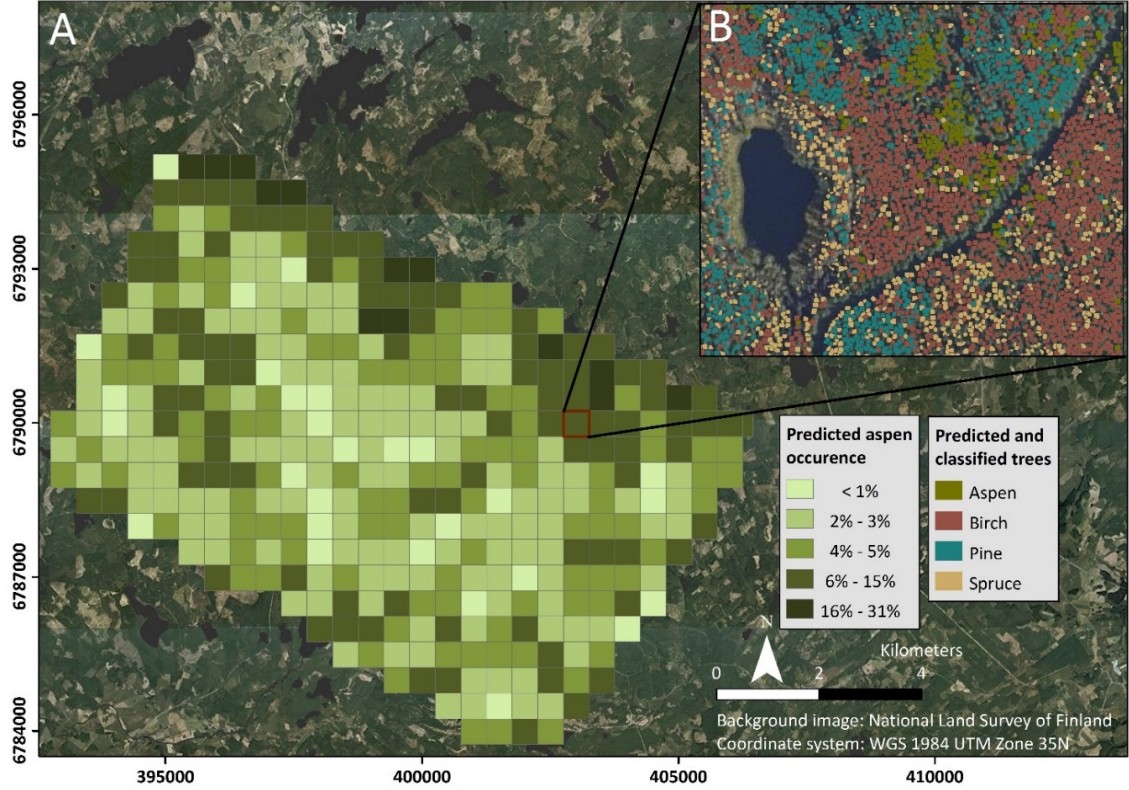

**Figure 12.** Percentage of the predicted aspen occurrence in 500 m grid based on the SVM model with reflectance and VI features (**A**). Illustration of the tree species segmentation and classification (**B**).

At the moment, data collected by multispectral sensors mounted on satellite platforms are the only option for operational large-scale wall-to-wall mapping of aspen occurrence and abundance, offering high cost-effectiveness, repeated observations, and standardized, validated processing methods. Although the lower spatial resolution (10–30 m) of the most suitable multispectral sensors

at the moment, Sentinel-2 and Landsat-8, makes single-tree classification practically impossible, the use of multitemporal Landsat-8 [103] or Sentinel-2 [104] data has given promising results in increasing classification accuracy of common tree species in boreal forests at stand level. However, the broader spectral bandwidths (low spectral resolution) of these sensors cannot capture the subtle differences in reflectance variations, which makes upscaling the results of this study difficult with only multispectral data.

In Finnish boreal forests, one possible solution for cost-effective tree species mapping on a larger scale would be to utilize data from a hyperspectral satellite and the ongoing nation-wide ALS data collection campaign (pulse density 5 p/m$^2$) coordinated by National Land Survey, while training and validation data could be derived from field-based forest inventory data. In a previous study, Ghosh et al. [40] confirmed that tree species maps produced from the now decommissioned Hyperion satellite at 30 m spatial resolution are often comparable with maps produced with airborne hyperspectral and LiDAR data. Expectations are, thus, high for the next generation of upcoming hyperspectral satellite missions, such as German Aerospace Center's (DLR) Environmental Mapping and Analysis Program EnMap [105] and National Aeronautics and Space Administration's (NASA) Hyperspectral Infrared Imager HyspIRI [106].

## 5. Conclusions

The aims of this research were to (i) compare the performance of hyperspectral data, vegetation indices, and their combination in tree species classification using the support vector machine and random forest classifiers, and to (ii) find the most important spectral features to discriminate aspen from the other common tree species. Regarding the first objective, support vector machine outperformed random forest in tree species classification, resulting in higher classification accuracies (both overall accuracy and Kappa values) when reflectance values, vegetation indices or their combination were used as a feature set. When principal component analysis was used as a feature set, there was no difference in the classification accuracies between the two classifiers. Regarding the second objective of the research, we noticed that by utilizing the hyperspectral data, it was possible to identify various spectral wavelengths and vegetation indices operating in near- and shortwave regions that do not only provide a possibility to discriminate deciduous trees from coniferous trees, but also enable the distinction of aspen from birch. The combination of reflectance values together with vegetation indices provided the highest accuracy of 92% for aspen discrimination from other most common tree species in the study area. Our results contribute to the remote sensing-based tree species detection and classification by providing important insights into the spectral features that differentiate between the tree species, especially aspen, in the study area. We found similarities but also notable differences in the model performance and important spectral regions compared with previous studies. Our finding can increase spatial knowledge of forest ecosystem composition and provides a possibility of detecting ecologically important aspen regardless of the sparse and scattered occurrence in northern boreal forests. This is important since aspen is contributing significantly to the biological diversity of boreal forest landscapes that are key factors in biodiversity management and conservation efforts in the area.

**Supplementary Materials:** The following are available online at http://www.mdpi.com/2072-4292/12/16/2610/s1, Table S1: Vegetation indices calculated and used in tree species classification, Table S2 Spectral variance of each principal component used in the analysis, Table S3: Kappa and overall accuracy (OA) results for the classification with different principal component sets using support vector machine, Table S4: Statistical significance of the difference in overall accuracy between the 15 models compared using McNemar's test (significant differences >3.84 indicated with *). McNemar's test statistic values are on the left side of the diagonal, p-values on the right side. The models were trained with Random Forest (RF) and Support Vector Machines (SVM), using all features with and without Recursive Feature Elimination (RFE) and fitted with different feature sets (Refl.—VNIR + SWIR reflectance, VI—Vegetation Indices, PCA—Principal Component Analysis).

**Author Contributions:** Conceptualization, A.V., T.K., P.V.; methodology, A.V., P.H.; software, A.V., P.H., J.M.; validation and formal analysis, A.V., P.H.; investigation, A.V., P.V., T.K., T.T., S.K.-S., S.K., L.P.; Writing—original draft preparation, A.V.; Writing—review and editing, A.V., P.H., S.K.-S., S.K., T.T., J.M., L.P., P.V., T.K.; visualization, A.V., P.H. All authors have read and agreed to the published version of the manuscript.

**Funding:** This research was funded by and supported by the Integrated Biodiversity Conservation and Carbon Sequestration in the Changing Environment (IBC-Carbon) (project number 312559), the Strategic Research Council, the Academy of Finland; e-shape has received funding from the European Union's Horizon 2020 research and innovation programme under grant agreement 820852 and Mapping and Assessment for Integrated ecosystem Accounting (MAIA) under grant agreement 817527; ENVECO (Eurostat Grants 2019): Finnish Ecosystem Observatory project funded by the Ministry of the Environment.

**Acknowledgments:** The authors wish to thank Rami Piiroinen from Sharper Shape Inc. who provided support for data processing. We also wish to acknowledge CSC–IT Center for Science, Finland, for generous computational resources and excellent user support. We also want to thank Natural Resources Institute Finland (LUKE) for the use of the Evo field data and Evo campus of the Häme University of Applied Science for providing accommodation during the fieldwork.

**Conflicts of Interest:** The authors declare no conflict of interest.

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
