# Peer review of "Detecting European Aspen (Populus tremula L.) in Boreal Forests Using Airborne Hyperspectral and Airborne Laser Scanning Data"

_remotesensing, doi:10.3390/rs12162610_

Round 1

Reviewer 1 Report

General comments:

The paper assesses the detection of the Aspen tree species in boreal forests using airborne hyperspectral and airborne laser scanning data and machine learning methods. Tree species detection is an important and hot topic for biodiversity conservation and I think the authors did a great job on this paper. Overall, the paper is well written, concise and clear, easy to read and understand. I think the paper is adequate for publishing in the Remote Sensing journal, and it only requires minor revision prior to publication. I could not find / raise many issues, as authors really did a nice job and produced a neat paper. See below one general comment and some specific comments:

1- Discussion seems to be missing comments regarding the impact and potential errors with the tree crown delineation which was the first step of the proposed method. Not sure if some additional results (even in supplementary materials) are required for this.

Specific comments:

L38, “F-score” -> “F1-score” (more occurrences along the manuscript)

L157, “dgital” typo

L201, Figure 4, the tree crowns are kind of similar when observing the Lidar CHM. I wonder if you can visually distinguish the Aspen crowns with a combination of spectral data, e.g. R-G-B. If yes one possibility here for a future work is to use a deep learning method. Please
read more references on “deep learning methods to detect the Aspen tree crowns”. Your results are already very good, but deep learning methods (e.g. Unet) have been proving to be better than the traditional machine learning algorithms. Another advantage of these methods is that you dismiss the need for the ITCD algorithm which also induce errors. Perhaps this kind of thought could be useful to add in the discussion for future works, but it is up to you.

L334, Figure 5, colors between aspen and spruce to be kind of similar, a bit confusing at a first glance, but it should be fine

L467, bold by mistake – I think

L572-574, what is the point on lowering the dimensionality?

L607, missing a point before “In general”

Reviewer 2 Report

The manuscript, "Detecting European Aspen (Populus tremula L.) in Boreal Forests Using Airborne Hyperspectral and Airborne Laser Scanning Data", is generally well-written and clearly presented. The methods and findings could be of interest to readers from many backgrounds including remote sensing, forest management, ecology and biodiversity.

The major concern I have is related to the cost of the datasets. Both airborne LiDAR and high resolution hyper-spectral data are relatively expensive, particularly when you apply it over large areas of forest lands. I suggest authors to write a paragraph mentioning cost efficiency. Also, is the gained accuracy worth the data acquisition and processing cost?

Pls consider using cm when you talk about DBH.

Overall very interesting article to read. 
